# Condensation of Benzyl Carbamate with Glyoxal in Polar Protic and Aprotic Solvents

**DOI:** 10.3390/molecules28227648

**Published:** 2023-11-17

**Authors:** Artyom E. Paromov

**Affiliations:** Laboratory for Chemistry of Nitrogen Compounds, Institute for Problems of Chemical and Energetic Technologies, Siberian Branch of the Russian Academy of Sciences (IPCET SB RAS), Biysk 659322, Russia; nitrochemistry@mail.ru

**Keywords:** caged compounds, condensation, domino reactions, nitrogen heterocycles, 2,4,6,8,10,12-hexanitro-2,4,6,8,10,12-hexaazaisowurtzitane

## Abstract

The synthesis of substituted 2,4,6,8,10,12-hexaazaisowurtzitane via direct condensation is challenging. The selection of starting ammonia derivatives is very limited. The important step in developing alternative synthetic routes to these compounds is to investigate their formation process in detail. Here, we examined an acid-catalyzed condensation between benzyl carbamate and glyoxal in a ratio of 2:1 in a range of polar protic and aprotic solvents, and discovered a new process occurring during the cascade condensation of glyoxal with ammonia derivatives as well as discovered several processes hindering the formation of caged compounds. More specifically, a cyclic compound, *N*,*N*′-bis(carbobenzoxy)-3,6-diamino-1,4-dioxane-2,5-diol, was found to form at the early stage of condensation under low acidity conditions. The formation of this compound is governed by an easier condensation of alcohol groups compared to the amide ones. The condensation intermediates, *N*,*N*′-bis(carbobenzoxy)ethan-1,2-diol, *N*,*N*′,*N*″-tris(carbobenzoxy)ethanol, and *N*,*N*′,*N*″,*N*‴-tetrakis(carbobenzoxy)ethan, were obtained at a higher acidity. A range of solvents were identified: those that react with benzyl carbamate, those that promote the progress of side processes, and those that promote precipitation of condensation intermediates. A few byproducts were isolated and identified. It was found that DMSO exhibits a strong deactivating ability, while CH_3_CN exhibits a strong activating ability towards the acid-catalyzed condensation process of benzyl carbamate with glyoxal.

## 1. Introduction

The search for and development of synthetic methods for new, more powerful high-energy-density materials (HEDMs) is a relevant objective for chemists and materials scientists worldwide [1,2,3,4,5,6,7,8,9,10,11,12,13]. One of the most actively developing directions in the synthesis of these substances is the synthesis of caged nitramines of aza- and oxaazaisowurtzitanes.

The first works that focused on the synthesis of caged nitramines incorporating oxaazaisowurtzitane cages appeared in the early 1980s but did not gain publicity [14,15]. The publicly available studies on the synthesis of the compact strained compound, 2,4,6,8,10,12-hexabenzyl-2,4,6,8,10,12-hexaazatetracyclo[5.5.0.0^3,11^.0^5,9^]dodecane (2,4,6,8,10,12-hexabenzyl-2,4,6,8,10,12-hexaazaisowurtzitane, HBIW), and on its nitration to 2,4,6,8,10,12-hexanitro-2,4,6,8,10,12-hexaazatetracyclo[5.5.0.0^3,11^.0^5,9^]dodecane (2,4,6,8,10,12-hexanitro-2,4,6,8,10,12-hexaazaisowurtzitane, CL-20) (Figure 1 and Figure 1) appeared in the mid-1990s only [16,17,18,19,20]. The energetic performance of CL-20 aroused an extreme interest of specialists engaged in the synthesis of HEDMs. CL-20 exhibits one of the highest detonation velocities (V_0_*D* = 9.36 (ε) km/s) among all explosives and the highest density among the known nitramines (ρ = 2.044 g/cm^3^) [18,19,21,22,23], as well as a positive enthalpy of formation (ΔH*f* = 454 kJ/mol), optimal oxygen balance (−11.0) and detonation pressure (420 kbar) [23,24,25,26,27,28]. CL-20 is superior to other high-energy explosive materials such as HMX, RDX, PETN, etc., CL-20 is touted as a potential component of solid propellants [29,30,31,32,33,34,35,36,37,38] and composite explosives [39,40,41,42,43,44,45,46,47,48,49,50,51,52,53,54,55].

At present, CL-20 is the most efficient HEDM among those being industrially produced. Active efforts are underway to find new synthetic methods and improve the known ones [15,23,56,57,58,59,60,61,62,63,64,65,66,67,68,69,70,71,72,73,74,75,76,77,78,79]. In addition, the possibility is being explored to better the properties of CL-20 through upgrading the molecule (replacement of nitro groups by other explosophoric groups; crosslinking with other high-energy bulky molecules) [80,81,82,83,84,85,86,87].

CL-20 is commercially produced by the multistage transfunctionalization of HBIW, which in turn is prepared by the condensation of benzylamine with glyoxal in about 89−95% yield [15,23,88,89,90]. All the commercial synthetic processes of CL-20 involve the catalytic reductive debenzylation of HBIW to 4,10-dibenzyl-2,6,8,12-tetraacetyl-2,4,6,8,10,12-hexaazatetracyclo[5.5.0.0^3,11^.0^5,9^]dodecane (4,10-dibenzyl-2,6,8,12-tetraacetyl-2,4,6,8,10,12-hexaazaisowurtzitane, TADBIW) (Figure 1) [15,23]. The reduction uses a Pd-containing catalyst in the presence of acetic anhydride and a bromine source [91]. Despite the high yields of TADBIW (80–90%) [92,93,94,95], this stage is the most considerable contributor to the cost of CL-20. The Pd catalyst is poisoned by the debenzylation products and requires frequent replacement [96,97].

The problem associated with the high cost of CL-20 substantially limits the application of this HEDM and can be settled in two ways: the first one is by upgrading the known synthetic methods and the second one is by developing conceptually new approaches to its synthesis. Despite researchers from all around the world mostly focusing their efforts on the first method for as many as several decades, they have failed to replace the debenzylation stage or appreciably reduce its cost. In this regard, the second method is growing more urgent.

The most promising approach to the synthesis of CL-20 is a two-stage synthesis involving a condensation stage that produces the 2,4,6,8,10,12-hexaazaisowurtzitane derivative and a stage of exhaustive nitration of the resultant cage to CL-20. The major challenge of this approach is that it is difficult to pick ammonia derivatives capable of furnishing the 2,4,6,8,10,12-hexaazaisowurtzitane derivative with easily nitratable N-substituents [14,15,23].

Hexaazaisowurtzitane derivatives can only be obtained in acceptable yields by the condensation of glyoxal with benzylamine (its derivatives), benzylamine-like compounds (a series of amines bearing an aromatic moiety linked via a methylene bridge to the amino group), allylamine (or similar compounds) and, as it turned out most recently, with a primary amine where the amino group is linked to the strained aliphatic ring such as cyclopropylamine and cyclobutylamine [15,23,98,99].

The key to solving the problems related to developing the two-stage synthetic method for CL-20 can only be a detailed study of the formation process of the 2,4,6,8,10,12-hexaazaisowurtzitane cage. Since HBIW is assumed to be formed as a result of trimerization of the corresponding diimine [17] and benzylamine is the best ammonia derivative for the synthesis of 2,4,6,8,10,12-hexaazaisowurtzitane, it is of interest to explore the condensation of glyoxal with structurally similar compounds that have the ability to form imines. The present study reports the results of an acid-catalyzed condensation between benzyl carbamate (**1**) and glyoxal in protic and aprotic solvents.

## 2. Results and Discussion

We previously investigated the condensation of benzamide [100] and some substituted sulfonamides [101,102,103,104] with glyoxal whereby a range of new oxaazaisowurtzitane derivatives was obtained and several process mechanisms and regularities were discovered. The present study proposes investigating the formation of aza- and oxaazaisowurtzitanes, as well as finding ammonia derivatives suitable for this process. Benzyl carbamate (**1**) was chosen herein as a new substrate for the study. This compound is similar in structure to benzylamine and is capable of yielding imines [105]. We failed to find information on the condensation between carbamic acid esters and glyoxal.

The acid-catalyzed condensation between carbamate **1** and glyoxal was carried out in a ratio of 2:1 in polar protic (H_2_O, HCOOH (FoOH) and AcOH) and aprotic solvents (AcOEt, Et_2_O, (CH_3_)_2_CO, CH_3_CN, CH_2_Cl_2_, THF and DMSO) at room temperature. The acid catalyst used was H_2_SO_4_. The reaction products were analyzed by HPLC. Preparative chromatography and recrystallization were employed to isolate pure compounds.

A few major reaction products were formed in the chosen solvents under moderate acidity conditions. All these products were isolated and identified: *N*,*N*′-bis(carbobenzoxy)-3,6-diamino-1,4-dioxane-2,5-diol (**2**); *N*,*N*′-bis(carbobenzoxy)ethan-1,2-diol (**3**); *N*,*N*′,*N*″-tris(carbobenzoxy)ethanol (**4**); and *N*,*N*′,*N*″,*N*‴-tetrakis(carbobenzoxy)ethan (**5**) (Figure 2). The greatest difficulty was to isolate compounds **2**–**4** that are unstable in acidic and alkaline media. They cannot be heated and held in solvents for a long time.

Compound **2** was probably formed as a result of the condensation of two *N*-carbobenzoxyethan-1,2,2-triol (**6**) molecules (Figure 3).

Three byproducts were isolated in addition to the condensation products of carbamate **1** with glyoxal. *N*,*N*′,*N*″-tris(carbobenzoxy)-2-ethoxyethan (**7**) was obtained in Et_2_O and AcOEt, which is a condensation product of compound **4** with EtOH (Figure 4). EtOH required for the formation of this compound was formed by the hydrolysis of the solvents in the reaction mixture.

The hydrolysis of carbamate **1** was found to proceed in Et_2_O, THF, AcOEt and CH_2_Cl_2_ when the content of H_2_SO_4_ in the mixture was 2% and higher, as is corroborated by the formation of *N*,*N*′,*N*″-tris(carbobenzoxy)-2-benzoxyethan (**8**), a condensation product of compound **4** and benzyl alcohol (BnOH) (Figure 5).

The condensation between carbamate **1** and glyoxal in FoOH did not almost take place. The major reaction product was a re-esterification product of benzyl formate (**9**) (Figure 6). The process proceeded actively even without additional acidification with FoOH.

We investigated the condensation of carbamate **1** with glyoxal in aqueous H_2_SO_4_ first. Because carbamate **1** is poorly water-soluble, its dissolution was performed in aqueous H_2_SO_4_, afterwards glyoxal was added portionwise to the reaction mixture with stirring. The process was carried out for 3 h. In all cases, the portionwise addition of glyoxal produced an abundant precipitate. When the time was over, the reaction mixture was diluted with H_2_O and filtered off.

Table 1 summarizes the composition of the principal reaction products of the acid-catalyzed condensation between carbamate **1** and glyoxal over H_2_SO_4_ of different concentrations. The syntheses were affected for 3 h.

As is seen in Table 1, the condensation process in aqueous H_2_SO_4_ stopped at compound **5** which was formed in a small quantity and precipitated from the reaction mixture alongside compounds **2**–**4**. Apart from the poor solubility of compounds **2**–**5** in aqueous H_2_SO_4_, the condensation was hindered by hydrolysis of the starting carbamate **1** through the ester group, as can be observed from the formation of compound **8**. The hydrolysis process accelerated as the acidity was raised (Table 1, Entries 1–5).

Next, we examined the condensation in the chosen organic solvents. Table 2 summarizes the composition of the major reaction products of the acid-catalyzed condensation between carbamate **1** and glyoxal in a range of polar protic and aprotic organic solvents over H_2_SO_4_ of different concentrations. The syntheses were effected for 20 h.

The analysis of data outlined in Table 2 allows for some conclusions. For instance, an obvious relationship is observed between the condensation process activity and the acidity of the medium. In weak organic acids (Table 2, Entries 18 and 21) and in most of the organic solvents in question when the content of H_2_SO_4_ in the mixture was 0.3% (Table 2, Entries 1, 3, 6, 8, 13 and 15), compounds **2**–**4** were formed first. The formation of compound **2** required a considerably lower acidity than that of compound **5**, indicating a higher condensation rate between the alcohol groups in the reaction medium compared to the condensation between the amide group of carbamate **1** and the alcohol groups. As the H_2_SO_4_ content in the mixture was raised to 2–7%, the condensation process was activated between compound **4** and carbamate **1** to give compound **5** in this case. This relationship is best visible in THF, DMSO and AcOH (Table 2, Entries 3–5, 15–17 and 21–23).

The main obstacles in the path of the acid-catalyzed condensation of carbamate **1** with glyoxal in the chosen solvents were side processes, low solubility of the condensation intermediates in the reaction mixture, and the deactivating effect of the solvent.

The side processes proceeded most actively in (CH_3_)_2_CO, AcOEt, FoOH, Et_2_O, CH_2_Cl_2_ and CH_3_CN.

In Et_2_O and AcOEt, the major byproduct was compound **7** (Table 2, Entries 1, 2, 6 and 7). The absence (a low yield) of the reaction products of EtOH with compound **3** can be suggestive of a low hydrolysis rate of the solvents under the reaction conditions. The formation of compound **3** was faster than the accumulation of EtOH in the reaction mixture. Compound **7** was formed in a considerable amount in unstable AcOEt at already 0.3% H_2_SO_4_, as opposed to Et_2_O (Table 2, Entry 6).

We noted previously that (CH_3_)_2_CO was readily engaged in a cascade of side condensation processes at an already small content of H_2_SO_4_ in the mixture [100]. However, the addition of 0.3–2% H_2_SO_4_ to the reaction mixture did not lead to an overly active formation of those compounds, and the condensation products of carbamate **1** and glyoxal were formed together with byproducts (Table 2, Entries 8–9). As the amount of H_2_SO_4_ was raised to 7%, the formation of byproducts accelerated so much that the reaction mass became resinified within 20 h, acquiring a dark-brown hue.

The condensation was slow in CH_2_Cl_2_ at 0.3% H_2_SO_4_ (Table 2, Entry 13). Only a small amount of compounds **2**–**5** was generated within 20 h. When the content of H_2_SO_4_ in the mixture was raised to 2%, the condensation process was activated and the hydrolysis process of carbamate **1** began concurrently to liberate BnOH (Table 2, Entry 14). Byproduct **8** was formed in 11% and compound **5** low soluble in CH_2_Cl_2_ precipitated in a great amount. The process performed at 7% H_2_SO_4_ in the reaction mixture resulted in a great quantity of byproducts, which were probably the condensation product of BnOH with carbamate **1** and the condensation products of BnOH with condensation intermediates of carbamate **1** and glyoxal.

Unlike (CH_3_)_2_CO, AcOEt, FoOH, Et_2_O and CH_2_Cl_2_, the condensation in CH_3_CN at 0.3–2% H_2_SO_4_ in the mixture took place actively without the formation of byproducts and hydrolysis products. However, when the acidity of the medium was raised to 7%, the amount of the resulting products **4** and **5** decreased abruptly, and products with a short retention time (HPLC), shorter than that of carbamate **1**, were observed to form in the stock solution. Probably at least one of these products was a reaction product (a short HPLC retention time) of acetonitrile because it was not formed in the other solvents.

The condensation intermediates were observed to precipitate actively in Et_2_O, CH_2_Cl_2_, CH_3_CN and AcOH, shifting the reaction equilibrium.

When the process was carried out in Et_2_O at 2% H_2_SO_4_ in the reaction mixture, a great amount of compound **4** was formed that precipitated from the reaction mixture. That said, no formation of compound **5** was noticed, which is likely suggestive of an extremely low solubility of compound **4** in this solvent (Table 2, Entry 2).

The condensation carried out in CH_2_Cl_2_, CH_3_CN and AcOH was accompanied by the precipitation of compound **5**. The greatest quantity of this compound within 20 h was formed in CH_2_Cl_2_ (Table 2, Entry 14) and CH_3_CN (Table 2, Entry 11) over 2% H_2_SO_4_ in the mixture, and in AcOH (Table 2, Entry 23) over 7% H_2_SO_4_.

When the process was performed in pure AcOH for 20 h, compound **2** precipitated in a small quantity (Table 2, Entry 21). AcOH appeared to possess sufficient acidity for the formation of compound **6** and its condensation to compound **2**.

It can be hypothesized that the activating effect of the polar aprotic solvents on the acid-catalyzed condensation in question will depend on their dipole moments but this dependence was not observed. Having similar dipole moments, solvents like CH_3_CN (3.92 D) and DMSO (3.96 D) exerted an opposite effect on the condensation process.

The condensation process was the slowest in DMSO. Even when the content of H_2_SO_4_ in the mixture was 7%, no active formation of condensation products was recorded (Table 2, Entry 17). DMSO behaves itself in a similar manner when benzylamine is condensed with glyoxal [100]. The ability of this solvent to deactivate the acid-catalyzed condensation process was employed by us in the analysis of unstable condensation intermediates **2**–**4**, which were dissolved in DMSO-D6 for ^1^H and ^13^C NMR analysis. It is quite probable that DMSO is likewise able to deactivate the condensation of other unstable compounds that structurally contain active hydroxyls.

The highest activating ability with respect to the condensation process was observed in CH_3_CN (Table 2, Entries 10–12). Compounds **4** and **5** were observed to form actively at already 0.3% H_2_SO_4_ in the mixture and began to precipitate from the reaction mixture when glyoxal was added portionwise.

The analysis of the HPLC data for the condensation products of carbamate **1** and glyoxal in the chosen solvents allows for the conclusion that the condensation process in some solvents does not stop at the precipitation of compounds **4** and **5** but proceeds “deeper” to produce a set of more complex condensation products having a longer retention time (HPLC). The compounds (about 20 in number) were identical for all the solvents and did not precipitate from the reaction mixture. The formation of these compounds occurred in Et_2_O and THF at 2–7% H_2_SO_4_ in the mixture, in CH_2_Cl_2_ and AcOH at 2% H_2_SO_4_, in CH_3_CN at 0.3–2% H_2_SO_4_, and negligibly in FoOH. These products were formed most actively at the lowest acidity in CH_3_CN.

## 3. Materials and Methods

The reagents were purchased from commercial suppliers and used as received, unless mentioned otherwise. Commercially available compounds were used without further purification, unless stated otherwise. Melting points were determined on a Stuart SMP30 melting point apparatus (Bibby Scientific Ltd., Staffordshire, UK). Infrared (IR) spectra were recorded on a Simex FT-801 Fourier transform infrared spectrometer (Simex, Novosibirsk, Russia) in KBr pellets or in a liquid film. ^1^H and ^13^C NMR spectra were recorded on a Bruker AV-400 instrument (Bruker Corporation, Billerica, MA, USA) at 400 MHz and 100 MHz. Chemical shifts are expressed in ppm (δ). Elemental analysis was performed on a ThermoFisher FlashEA 1112 elemental analyzer (ThermoFisher, Waltham, MA, USA). For preparative chromatography, silica gel Kieselgel 60 (0.063–0.2 mm, Macherey-Nagel GmbH & Co. KG, Dueren, Germany) was used. HPLC analysis was performed on an Agilent 1200 chromatograph (Agilent Technologies, Waldbronn, Germany) with a diode array detector. The separation was carried out on a Hypersil ODS (100 × 2.1 mm, a 5 µm mesh) column. Mixed solvents A (0.2% phosphoric acid) and B (acetonitrile) were used as the mobile phase. The mobile phase composition was varied in the gradient mode: the concentration of solvent B was linearly raised from 20 to 50% for 5 min, then from 50 to 100% for 5 to 70 min, and maintained at this level for another 10 min. The flow rate of the eluent was 0.25 mL/min, the column temperature was 25 °C, the detection was run at a 210-nm wavelength, and the sample volume was 3 µL. The column conditioning time between successive injections was 15 min.

## 4. Experimental

### 4.1. N,N′-Bis(carbobenzoxy)-3,6-diamino-1,4-dioxane-2,5-diol (**2**)

Carbamate **1** (3 g, 0.020 mol) was added to AcOH (84 mL) with vigorous stirring. Next, after the carbamate was dissolved, glyoxal (1.44 g, 40%, 0.010 mol) was added to the reaction mixture, and the whole was allowed to be stirred at room temperature for 60 h and then filtered. The filter cake was washed with (CH_3_)_2_CO (4 × 5 mL), transferred to a beaker and muddled in (CH_3_)_2_CO (30 mL) at reflux for 20 min, and then filtered. The filter cake was washed with (CH_3_)_2_CO (2 × 6 mL) and dried to constant weight at room temperature (0.44 g; containing 77% of compound **2**). The dry sediment was further dissolved in a minimum quantity of DMSO at 60–70 °C, diluted twofold with acetonitrile and allowed to crystallize at room temperature for 24 h. Upon time completion, the mixture was filtered and the filter cake was washed with (CH_3_)_2_CO (3 × 5 mL) and dried to constant weight at room temperature to furnish compound **2** as a white powder.

Yield: 0.104 g (94.8% assay HPLC), 0.236 mmol (4.7% calculated on a glyoxal basis). Mp = 189–192 °C. IR (KBr): ν = 3369, 3311, 3060, 3029, 2969, 2895, 1699, 1525, 1452, 1360, 1343, 1243, 1145, 1086, 1044, 986, 964, 911, 881, 856, 784, 745, 697, 662 cm^−1^. ^1^H NMR (DMSO-D6): δ = 4.84 (s, 2H), 5.05 (q, *J_1_* = 12.9, *J_2_* = 17.1, Hz, 4H), 5.41 (d, *J* = 9.7 Hz, 2H), 6.99 (d, *J* = 2.5 Hz, 2H), 7.25 (d, *J* = 9.6 Hz, 2H), 7.35–7.32 (m, 10H) ppm. ^13^C{1H} NMR (DMSO-D6): δ = 66.0, 72.2, 90.0, 128.1, 128.3, 128.8, 137.3, 155.9 ppm. Elemental analysis, calcd (%) for C_20_H_22_N_2_O_8_ (418.40): C, 57.41; H, 5.30; N, 6.70; O, 30.59; found: C, 55.84; H, 5.22; N, 6.58; O, 29.63 (see Appendix A).

### 4.2. N,N′-Bis(carbobenzoxy)ethan-1,2-diol (**3**)

H_2_SO_4_ (0.202 mL, 0.37 g, 94%) was added to Et_2_O (80 mL) with vigorous stirring and cooling with ice water. The reaction mixture was then heated to 15–20 °C; afterwards, carbamate **1** (4 g, 0.026 mol) was added thereto with stirring. After the carbamate was dissolved, glyoxal (1.92 g, 40%, 0.013 mol) was added portionwise to the mixture for 10 min at 22–25 °C. Once the portionwise addition was completed, the mixture was left to stir at room temperature for 45 min and then filtered. The filter cake was washed with Et_2_O (3 × 5 mL) and H_2_O (4 × 7 mL), transferred to a beaker and muddled in H_2_O (40 mL) for 1 h at room temperature, and then filtered. The filter cake was washed with H_2_O (3 × 7 mL) and dried to constant weight at room temperature (0.823 g; containing 53% of compound **3**). The resultant sediment was then muddled in (CH_3_)_2_CO (150 mL) for 1.5 h and then filtered. The filtrate was evaporated in a rotary evaporator at a bath temperature that was not above 25 °C. The residue was recrystallized from (CH_3_)_2_CO (without heating), and the filter cake was washed with Et_2_O (2 × 5 mL) and dried at room temperature to constant weight to give compound **3** as a white crystalline powder.

Yield: 0.298 g (95.4% assay HPLC), 0.827 mmol (6.0% calculated on a compound **1** basis). Mp = 147–149 °C. IR (KBr): ν = 3307, 3273, 3061, 3035, 2942, 1688, 1544, 1497, 1451, 1409, 1380, 1312, 1282, 1236, 1156, 1045, 1010, 968, 910, 779, 731, 694 cm^−1^. ^1^H NMR (DMSO-D6): δ = 4.88 (t, *J* = 5.1 Hz, 2H), 5.03 (q, *J*_1_ = 12.6, *J*_2_ = 19.1, 4H), 5.73 (br. s, 2H), 7.30–7.37 (m, 10H), 7.46 (d, *J* = 5.2 Hz, 2H) ppm. ^13^C{1H} NMR (DMSO-D6): δ = 65.7, 77.0, 128.2, 128.8, 137.5, 156.0 ppm. Elemental analysis, calcd (%) for C_18_H_20_N_2_O_6_ (360.36): C, 59.99; H, 5.59; N, 7.77; O, 26.64; found: C, 59.96; H, 5.62; N, 7.72; O, 26.56 (see Appendix A).

### 4.3. N,N′,N″-Tris(carbobenzoxy)ethanol (**4**)

H_2_SO_4_ (0.150 mL, 0.275 g, 94%) was added to Et_2_O (20 mL) with vigorous stirring and cooling with ice water. The reaction mixture was then heated to 15–20 °C; afterwards, carbamate **1** (0.717 g, 4.743 mmol) was added to the mixture with stirring. After the carbamate was dissolved, glyoxal (0.344 g, 40%, 2.370 mmol) was added portionwise to the mixture for 10 min at 22–25 °C. Once the portionwise addition was completed, the mixture was left to stir at room temperature for 20 h and then filtered. The filter cake was washed with Et_2_O (3 × 5 mL) and H_2_O (4 × 7 mL), transferred to a beaker and muddled in H_2_O (40 mL) for 1 h at room temperature, and then filtered. The filter cake was washed with H_2_O (3 × 7 mL) and dried to constant weight at room temperature (0.152 g; containing 78% of compound **4**). The resultant sediment was then muddled in (CH_3_)_2_CO (15 mL) at room temperature for 1.5 h and then filtered. The filtrate was evaporated in a rotary evaporator at a bath temperature that was not above 25 °C. The resultant sediment was dissolved in a minimum quantity of (CH_3_)_2_CO (without heating); afterwards, compound **4** was crystallized out with hexane. For this, the solution had hexane added drop by drop until active crystallization started and was left to crystallize at room temperature for 24 h. Upon time completion, the suspension was filtered, and the filter cake was washed with Et_2_O (2 × 8 mL) and dried to constant weight at room temperature to afford compound **4** as a white crystalline powder.

Yield: 0.067 g (94.0% assay HPLC), 0.128 mmol (8.1% calculated on a compound **1** basis). Mp = 175–177 °C. IR (KBr): ν = 3436, 3307, 3061, 3032, 2951, 2891, 1693, 1662, 1536, 1513, 1454, 1381, 1337, 1295, 1225, 1156, 1086, 1016, 967, 913, 844, 782, 737, 695 cm^−1^. ^1^H NMR (DMSO-D6): δ = 4.99–5.08 (m, 6H), 5.14 (br. s, 2H), 6.10 (s, 1H), 7.28–7.44 (m, 17H), 7.61 (d, *J* = 6.6 Hz, 1H) ppm. ^13^C{1H} NMR (DMSO-D6): δ = 62.8, 65.9, 66.0, 75.6, 128.2, 128.8, 137.4, 155.6, 155.9, 156.1 ppm. Elemental analysis, calcd (%) for C_26_H_27_N_3_O_7_ (493.508): C, 63.28; H, 5.51; N, 8.51; O, 22.69; found: C, 62.85; H, 5.54; N, 8.44; O, 22.39 (see Appendix A).

### 4.4. N,N′,N″,N‴-Tetrakis(carbobenzoxy)ethan (**5**)

H_2_SO_4_ (0.819 mL, 1.5 g, 94%) was added to AcOH (20 mL) with vigorous stirring and cooling with ice water. The reaction mixture was then heated to 15–20 °C; afterwards, carbamate **1** (0.717 g, 4.743 mmol) was added to the mixture with stirring. After the carbamate was dissolved, glyoxal (0.344 g, 40%, 2.370 mmol) was added portionwise to the mixture for 10 min at 22–25 °C. Once the portionwise addition was completed, the mixture was left to stir at room temperature for 20 h and then filtered. The filter cake was washed with AcOH (2 × 4 mL) and Et_2_O (3 × 5 mL) and dried to constant weight at room temperature (0.211 g; containing 82% of compound **5**). The sediment was transferred to a beaker and muddled in DMF (20 mL) at room temperature for 1 h, and then filtered. The filter cake was washed with Et_2_O (2 × 5 mL) and dried to constant weight at room temperature. The resultant sediment was then dissolved in a minimum quantity of hot DMSO at 50–60 °C, afterwards compound **5** was crystallized out with acetonitrile. For this, the solution was cooled down to room temperature, diluted eightfold with CH_3_CN and then allowed to undergo crystallization at room temperature for 24 h. Upon time completion, the suspension was filtered, and the filter cake was washed with Et_2_O (3 × 5 mL) and dried to constant weight at room temperature to furnish compound **5** as a white crystalline powder.

Yield: 0.125 g (96.0% assay HPLC), 0.191 mmol (16.1% calculated on a compound **1** basis). Mp = 278–280 °C. IR (KBr): ν = 3305, 3091, 3034, 2951, 2897, 1698, 1547, 1518, 1454, 1339, 1298, 1238, 1139, 1012, 981, 914, 844, 785, 757, 740, 697, 673 cm^−1^. ^1^H NMR (DMSO-D6): δ = 5.02 (s, 8H), 5.32 (br. s, 2H), 7.34 (s, 20H), 5.32 (br. s, 4H) ppm. ^13^C{1H} NMR (DMSO-D6): δ = 61.6, 66.0, 128.2, 128.8, 137.2, 155.7 ppm. Elemental analysis, calcd (%) for C_34_H_34_N_4_O_8_ (626.65): C, 65.17; H, 5.47; N, 8.94; O, 20.43; found: C, 64.43; H, 5.43; N, 8.85; O, 20.37 (see Appendix A).

### 4.5. N,N′,N″-Tris(carbobenzoxy)-2-ethoxyethan (**7**)

H_2_SO_4_ (0.218 mL, 0.4 g, 94%) was added to AcOH (20 mL) with vigorous stirring and cooling with ice water. The reaction mixture was then heated to 15–20 °C; afterwards, carbamate **1** (0.717 g, 4.743 mmol) was added to the mixture with stirring. After the carbamate was dissolved, glyoxal (0.344 g, 40%, 2.370 mmol) was added portionwise to the mixture for 10 min at 22–25 °C. Once the portionwise addition was completed, the mixture was left to stir at room temperature for 20 h and then filtered. The filter cake was washed with AcOH (2 × 5 mL) and Et_2_O (4 × 5 mL) and dried to constant weight at room temperature (0.255 g; containing 61% of compound **7**). The resultant sediment was then muddled in (CH_3_)_2_CO (15 mL) at room temperature for 1.5 h and then filtered. The filtrate was evaporated in a rotary evaporator at a bath temperature that was not above 30 °C. The residue was recrystallized from (CH_3_)_2_CO and dried to constant weight at room temperature to give compound **7** as a white crystalline powder.

Yield: 0.166 g (97.1% assay HPLC), 0.309 mmol (19.5% calculated on a compound **1** basis. Mp = 194–196 °C. IR (KBr): ν = 3308, 3062, 3032, 2970, 2927, 2897, 1692, 1535, 1514, 1454, 1380, 1337, 1297, 1274, 1162, 1099, 1052, 1017, 910, 843, 780, 737, 694 cm^−1^. ^1^H NMR (DMSO-D6): δ = 1.06 (t, *J* = 6.4 Hz, 3H), 3.36–3.55 (m, 2H), 4.97–5.10 (m, 7H), 5.22 (br. s, 1H), 7.35 (s, 15H), 7.53 (d, *J* = 6.7 Hz, 1H), 7.64 (d, *J* = 5.6 Hz, 1H), 7.76 (d, *J* = 9.0 Hz, 1H) ppm. ^13^C{1H} NMR (DMSO-D6): δ = 15.4, 61.6, 63.1, 65.84, 65.98, 66.05, 81.66, 128.14, 128.27, 128.33, 128.79, 128.82, 137.2, 137.4, 155.6, 155.8, 156.7 ppm. Elemental analysis, calcd (%) for C_28_H_31_N_3_O_7_ (521.56): C, 64.48; H, 5.99; N, 8.06; O, 21.47; found: C, 64.47; H, 6.02; N, 8.00; O, 21.33 (see Appendix A).

### 4.6. N,N′,N″-Tris(carbobenzoxy)-2-benzoxyethan (**8**)

H_2_SO_4_ (1.8 mL, 3.3 g, 94%) was added to H_2_O (2 mL) with vigorous stirring and cooling with ice water. The resultant mixture was then heated to room temperature and carbamate **1** (0.358 g, 2.368 mmol) was added with stirring. After the carbamate was dissolved, glyoxal (0.172 g, 40%, 1.185 mmol) was added portionwise to the mixture for 10 min at 22–25 °C. Once the portionwise addition was completed, the mixture was allowed to stir at room temperature for 3 h. Upon time completion, the reaction mixture was diluted with ice water (30 mL), stirred for 10 min and filtered. The filter cake was washed with H_2_O (5 × 7 mL) and dried to constant weight at room temperature (0.322 g; containing 25.1% of compound **8**). The resultant sediment was separated by preparative chromatography. Mixed CHCl_3_:CH_3_CN:AcOH in a ratio of 10:1:0.15 *v*/*v* was used as the eluent. Some of the sediment (compound **7**) was insoluble in the eluent and should be filtered off prior to preparative chromatography. A fraction comprising at least 85% of compound **8** (Rf 0.69; HPLC control) was collected. Chloroform and acetonitrile were evaporated from the collected fraction in a rotary evaporator at a bath temperature not above 35 °C. Most of the AcOH was evaporated from the recovery flask with rubber bellow at room temperature. The resultant sediment with acetic acid traces was recrystallized from the same eluent as used in preparative chromatography, and the filter cake was washed with Et_2_O (2 × 5 mL) and dried to constant weight at room temperature to furnish compound **8** as a white powder.

Yield: 0.044 g (98.5% assay HPLC), 0.074 mmol (12.5% calculated on a compound **1** basis with allowance for hydrolysis of compound **1** to benzyl alcohol). Mp = 201–203 °C. IR (KBr): ν = 3320, 3089, 3062, 3034, 2959, 2882, 1710, 1693, 1556, 1530, 1514, 1453, 1379, 1341, 1297, 1270, 1238, 1150, 1101, 1050, 1023, 997, 973, 910, 825, 775, 754, 731, 695, 676 cm^−1^. ^1^H NMR (DMSO-D6): δ = 4.51 (q, *J*_1_ = 11.8, *J*_2_ = 36.6, 4H), 5.04–5.13 (m, 7H), 5.32 (br. s, 1H), 7.29–7.34 (m, 20H), 7.66 (dd, *J*_1_ = 6.7, *J*_2_ = 27.3, 2H) 7.95 (d, *J* = 9.0 Hz, 1H) ppm. ^13^C{1H} NMR (DMSO-D6): δ = 61.6, 65.90, 65.99, 66,13, 69.3, 81.9, 127.82, 127.96, 128.16, 128.24, 128.33, 128.6, 128.80, 128.82, 137.24, 137.29, 137.33, 138.5, 155.6, 155.8, 156.8 ppm. Elemental analysis, calcd (%) for C_33_H_33_N_3_O_7_ (583.63): C, 67.91; H, 5.70; N, 7.20; O, 19.19; found: C, 67.85; H, 5.76; N, 7.15; O, 19.10 (see Appendix A).

## 5. Conclusions

The acid-catalyzed condensation between benzyl carbamate and glyoxal in a molar ratio of 2:1 was investigated in detail in a range of polar protic and aprotic solvents. Benzyl carbamate proved to be an active amide sufficiently stable towards acid hydrolysis and had the ability to engage in a condensation reaction with glyoxal under relatively low acidity conditions. For instance, in a relatively weak AcOH, the condensation between two benzyl carbamate molecules and two glyoxal molecules proceeded to yield a cyclic compound, *N*,*N*′-bis(carbobenzoxy)-3,6-diamino-1,4-dioxane-2,5-diol (**2**). Increasing the acidity of the solvents by adding H_2_SO_4_ to the reaction mixture allowed the activation of the condensation process and the preparation of products containing a greater quantity of secondary amido groups: *N*,*N*′-bis(carbobenzoxy)ethan-1,2-diol (**3**), *N*,*N*′,*N*″-tris(carbobenzoxy)ethanol (**4**) and *N*,*N*′,*N*″,*N*‴-tetrakis(carbobenzoxy)ethan (**5**).

The main obstacles in the path of the acid-catalyzed condensation of benzyl carbamate with glyoxal were discovered herein, which are side processes, the low solubility of condensation intermediates in the reaction mixture, and the deactivating effect of the solvent.

Side processes were noted to take place in most solvents under study and their rate depended on the acidity of the medium. *N*,*N*′,*N*″-tris(carbobenzoxy)-2-ethoxyethan (**7**) was formed in AcOEt and Et_2_O, re-esterification of benzyl carbamate took place actively in FoOH to give benzyl formate (**9**), while a range of byproducts was formed in (CH_3_)_2_CO by the transformation of (CH_3_)_2_CO. A great many byproducts with a short retention time (HPLC) were observed to form in solvents such as CH_2_Cl_2_ and CH_3_CN when the content of H_2_SO_4_ in the mixture was 7%. Hydrolysis of benzyl carbamate took place in H_2_O and CH_2_Cl_2_ to furnish a byproduct, *N*,*N*′,*N*″-tris(carbobenzoxy)-2-benzoxyethan (**8**).

The condensation intermediates were observed to precipitate actively in Et_2_O, CH_2_Cl_2_, CH_3_CN and AcOH, shifting the reaction equilibrium.

It has been established that there is no direct correlation between the dipole moments of the polar aprotic solvents and the activating effect with respect to the acid-catalyzed condensation process of benzyl carbamate with glyoxal.

DMSO had the strongest deactivating effect on the acid-catalyzed condensation process of benzyl carbamate with glyoxal. This feature of DMSO can probably be utilized for the stabilization of other compounds bearing active hydroxyls.

It can thus be concluded that H_2_O, Et_2_O, CH_2_Cl_2_, AcOEt, (CH_3_)_2_CO and DMSO are the worst solvents for the cascade condensation of benzyl carbamate with glyoxal. The use of these solvents for an acid-catalyzed condensation of other ammonia derivatives having a similar or lower basicity will probably be difficult as well.

Among the solvents chosen, CH_3_CN had the strongest activating effect on the acid-catalyzed condensation process between benzyl carbamate and glyoxal. The addition of as low as 0.3% H_2_SO_4_ to the reaction mixture is enough for an active formation of *N*,*N*′,*N*″,*N*‴-tetrakis(carbobenzoxy)ethan (**5**) and a range of more complex condensation products having a longer retention time than compound **5** (HPLC). The high activating ability of CH_3_CN with respect to the acid-catalyzed condensation reaction is likely a major factor that makes this solvent the best for the synthesis of 2,4,6,8,10,12-hexaazaisowurtzitane, which is particularly corroborated by the highest yields of HBIW and other hexaazaisowurtzitanes in this solvent.

## Data Availability

Data are contained within the article and Appendix A.

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
