# Peer review of "Condensation of Benzyl Carbamate with Glyoxal in Polar Protic and Aprotic Solvents"

_molecules, 2023, doi:10.3390/molecules28227648_

Round 1
Reviewer 1 Report
Comments and Suggestions for Authors
In the study under review, the author addresses the challenging problem of the synthesis of caged compounds of aza- and oxaazaisowurtzitanes—the precursors of promising materials with a high energy density (HEDMs). The said problem is associated with the formation process of the 2,4,6,8,10,12-hexaazaisowurtzitane cage being highly selective towards ammonia derivatives. The main focus of the manuscript is on investigating the formation processes and finding synthetic conditions of 2,4,6,8,10,12-hexaazaisowurtzitane derivatives as the precursors of the most efficient and domesticated HEDM—2,4,6,8,10,12-hexanitro-2,4,6,8,10,12-hexaazaisowurtzitane (CL-20). Today, the synthesis of CL-20 is a high-cost multistage process.
The Results and Discussion section demonstrates that the author is well-qualified in the field of organic chemistry. The author presented research data on the acid-catalyzed condensation reaction between benzyl carbamate and glyoxal in a ratio of 2:1 performed in a series of polar protic and aprotic solvents, and discovered a new process occurring in the cascade condensation of glyoxal with ammonia derivatives, as well as a few processes preventing the caged compounds from formation. In particular, this study has shown for the first time that 3,6-diamino-1,4-dioxane-2,5-diol derivatives are formed at the early stage of the condensation between glyoxal and ammonia derivatives and that there is no direct correlation between dipole moments of the solvents and the ability of the solvents to activate the condensation process. Moreover, the study identified the least and most suitable solvents for the condensation of glyoxal with benzyl carbamate and, as the author suggests, with ammonia derivatives having a similar basicity. Acetonitrile was noted as the best solvent for finding the formation conditions of 2,4,6,8,10,12-hexaazaisowurtzitane derivatives.
The obtained results are original and will be of interest to organic synthesis experts, especially to those who address the problems of the synthesis of caged derivatives of aza- and oxaazaisowurtzitanes.
The overall work is interesting and well-written, so I recommend publication after minor revision.
Specific comments:
In the Conclusions section, the author lists the least appropriate solvents for the condensation of benzyl carbamate with glyoxal, among which water is missing. Given the text of the manuscript, water is a poor solvent for the process as well. Please add water to that list or explain why it is missing on the list.
Reviewer 2 Report
Comments and Suggestions for Authors
This paper contains a detailed account of a focussed research project aimed at understanding the formation of a complex heterocycle, and expanding the utility of the synthetic approach. Although a known synthesis, there is scope for improvement on the reported method, and the work in this paper helps us to understand more about controlling the synthesis. Of course, primary carbamates are very different to the primary amines normally used, as they are much less basic, however some interesting results were obtained. The paper will be of some general interest but of specific interest to readers of the journal. The paper is well-written and the results are well explained. There is a good level of practical detail and some useful supporting information.
Whether or not it is possible to find an efficient route to a complex target molecule which improves upon the more complex one currently used. The topic is quite focussed but very important in its field. The new methodology may have broader application in synthesis beyond the specific application in the paper. A new potential route to a complex target molecule. It seems quite comprehensive, and very interesting. The Figures and Tables serve to communicate the results well.
